# Combining Influenza and COVID-19 Booster Vaccination Strategy to Improve Vaccination Uptake Necessary for Managing the Health Pandemic: A Systematic Review and Meta-Analysis

**DOI:** 10.3390/vaccines11010016

**Published:** 2022-12-21

**Authors:** Nikolaos Tzenios, Mary E. Tazanios, Mohamed Chahine

**Affiliations:** 1Public Health and Medical Research, Charisma University, Grace Bay TKCA 1ZZ, Turks and Caicos Islands; 2Doctor of Health Sciences Candidate Program, MCPHS University, Boston, MA 02115, USA; 3Harvard Medical School Postgraduate Medical Education High Impact Cancer Research 2019–2021, Boston, MA 02115, USA; 4Clinical Research, TRG GEN+, Beirut 0000, Lebanon; 5Biological and Chemical Technology, International Medical Institute, Kursk State Medical University, 305030 Kursk, Russia

**Keywords:** COVID-19 booster, influenza vaccine, vaccination strategy, combining vaccination

## Abstract

**Background:** The uptake of COVID-19 booster vaccines has been significantly low. Therefore, it is questionable whether combining the COVID-19 booster vaccines with influenza vaccines can increase the population’s interest in taking such vaccines and manage the health pandemic effectively. **Methodology:** In this systematic review and meta-analysis, a synthesis of the findings and summary of a total of 30 research articles based on the topic, ‘combining influenza and COVID-19 booster vaccination strategy’ was undertaken. The research articles were identified from three databases, namely, PubMed, Cochran Library, and Google Scholar using specific keywords and inclusion criteria. However, research articles that were not peer-reviewed and not published in English were excluded from the systematic review and meta-analysis. The average risk ratio of the intervention group getting a combination of COVID-19 booster and influenza vaccines from the samples of the included studies was 0.78 with regard to a 95% CI. Such risk ratio is based on the null hypothesis of the current study that combining COVID-19 booster and influenza vaccines can increase the uptake of COVID-19 booster vaccines. On the other hand, the heterogeneity between such studies was I^2^ = 35%, while the statistical significance of their findings occurred at *p* < 0.05. The average *p*-value of the included research studies was *p* = 0.62 with the proportion of studies with significant *p*-values being 63.33% which is equivalent to 19 out of 30 studies. Therefore, the null hypothesis was not rejected in more than half of the studies. **Results:** A synthesis of the chosen research articles revealed that when influenza and COVID-19 booster vaccines are combined, there is potential for an increase in the uptake of the latter, mainly because many populations have already been accustomed to taking influenza vaccines on an annual basis. **Conclusions:** In this way, through such findings, medical health experts can make informed decisions to increase the population’s willingness to receive the COVID-19 booster vaccines.

## 1. Introduction

Despite the fact that medical scientists consider COVID-19 vaccines to be effective in managing the health effects associated with the virus, the interest by different populations to take such vaccines has been very low. On that account, researchers within the field of medicine and health sciences have questioned whether combining influenza vaccines with COVID-19 booster vaccines, illustrated in Figure 1, can increase the latter’s uptake. The main rationale behind their hypotheses is that the number of people receiving influenza vaccines exceeds those who have had an interest in taking different COVID-19 vaccines. Previous systematic reviews and meta-analysis have only focused on synthesizing and summarizing findings of research articles related to combining other vaccines such as chickenpox, hepatitis A and B, HPV vaccines, among others. However, the current study involves a systematic review and a meta-analysis of some research articles that have focused on combining influenza and COVID-19 booster vaccination to increase vaccine uptake among populations adversely affected by the health pandemic. On that note, this systematic review and meta-analysis holds value due to the fact that it will be the first to synthesize and summarize the findings of studies that have focused on researching on the combination of COVID-19 booster and influenza vaccines. It will focus on the null hypothesis that a combination of COVID-19 booster and influenza vaccines can be an effective strategy in increasing the uptake of the COVID-19 booster vaccine. It is the appropriate time to undertake such systematic review and meta-analysis mainly because of the need to make informed health decisions on mitigating the health implications associated with such viruses, especially among diverse populations that are majorly affected by them. In this regard, the current systematic review and meta-analysis will play an effective role in enhancing the process of making sound public health decisions for the roll-out of COVID-19 vaccines, thereby decreasing the negative health effects that such virus is associated with. To achieve such goal, this study will aim to provide a synthesis and an overall summary of existing data collected using specific research methodologies by various researchers on the topic mentioned above. Since many populations have felt accustomed to receiving influenza vaccines on an annual basis, combining such vaccines with COVID-19 booster vaccines can be an effective strategy for increasing their interest in receiving the latter.

## 2. Survey Methodology

### 2.1. Search Strategy and Selection Criteria

The current study is a systematic review and a meta-analysis. As a systematic review, it aims to identify, evaluate, and summarize the findings of various previously conducted research studies on combining influenza and COVID-19 booster vaccination to increase the uptake of the latter by populations. By achieving such a goal, the current study will make available evidence related to the topic more accessible to healthcare decision-makers who may use such evidence to develop effective strategies for increasing the interest of populations in getting vaccinated against COVID-19. On the other hand, as a meta-analysis, the current study aims at assessing the results of previous research studies related to the topic stated above to provide an overall summary estimate from existing data on such topic and contribute towards its body of knowledge with well-informed conclusions. On that note, the current study will be based on the evaluation of randomized, controlled clinical trials to achieve the efficacy of its assessment of existing results of the research studies conducted on the topic of combining influenza and COVID-19 booster vaccination.

The current systematic review and meta-analysis involved specific inclusion criteria for research studies to ensure that only relevant studies were used. First, only studies that were conducted between 1 November 2020, and 1 November 2022, were used for the systematic review and meta-analysis. In essence, only research articles published within the last two years were considered for this study since its topic focused on COVID-19 booster vaccination, which has been one of the main discussed health topics during such period. Second, only research articles written and published in English were used for this study. Third, the current study involved a systematic review and meta-analysis of research studies that assessed populations that had shown reluctance to take COVID-19 booster vaccination. Lastly, the current systematic review and meta-analysis only included studies that had used randomized controlled trials as their study design while seeking to establish accurate and valid findings on the research topic. The study participants were all adults above 18 years old and had taken either the first or second dose of specific COVID-19 vaccines. The intervention used in some of the studies was providing education on COVID-19 vaccines and vaccination processes. The comparators assessed were the level of interest in taking influenza and COVID-19 vaccines. Lastly, the setting used for the analyzed studies was healthcare facilities only and the time points were between November 2020 and November 2022. Due to the current nature of the period in which the studies were conducted, there is a high level of confidence in the credibility, validity, and accuracy of their synthesized research findings.

There are two main reasons why some research studies on combining influenza and COVID-19 booster vaccination strategy were excluded from the current systematic review and meta-analysis. One such reason was that the studies might have only had abstracts and not full texts, which would have made it difficult to synthesize their research findings. The other reason was that the studies may not have been peer-reviewed or authored by experts within the fields of medicine, nursing, and health sciences. On that note, such studies were regarded to be having less accurate and valid research findings. Three databases were used to search for the articles used, as shown in Figure 2 below. They include PubMed, Google Scholar, and the Cochrane Library. The exact date cutoffs used during the search were 1 November 2020, and 1 November 2022. The search terms used in all the databases include ‘COVID-19 booster vaccination’, ‘influenza and COVID-19 booster vaccines’, ‘low COVID-19 booster vaccine uptake’, and ‘increasing COVID-19 booster vaccine uptake’ as shown in Appendix A. The thorough nature of the search was attributed to the fact that grey literature sources such as white papers, podcasts, blogs on the topic, and government reports were also assessed. Nonetheless, information from such sources was considered reliable and useful for the systematic review and meta-analysis, but none of such sources were included since they were not peer-reviewed in accordance with the exclusion criteria. No search was conducted on trial registries, implying that data were not obtained from unpublished studies. The data sought during the synthesis were individual patient-level since it was regarded as more accurate and credible in making conclusions about the current research topic than summary estimates. The database searches for the articles and the data extraction process were undertaken by the current study’s authors. An independent third party with sufficient knowledge on the topic of COVID-19 vaccines was responsible for solving conflicts of interest that arose due to the inclusion of the research articles. The inclusion criteria of the studies for the systematic review and that of the meta-analysis were the same and involved peer-reviewed research studies conducted and published within the last two years, which had been written in the English language only. Lastly, a link to the study protocol is yet to be available online.

### 2.2. Data Analysis

A logical method of data extraction was used to extract data for the systematic review and meta-analysis from the chosen relevant research articles. The specific type of logical method of data extraction that was used was full extraction, in which all data were extracted from the source research article directly at once before conducting another full extraction from another research article. To deal with duplicate data, the level of a match on the data was first assessed before removing the duplicated data that appeared less accurate based on the findings of other studies. The authors of this study also conducted the data appraisal and extraction. The method used for data appraisal was content analysis, in which similarities in the information within the study designs, data collection processes, collected data, and interpretation of findings were established to derive a similar conclusion on the topic of interest. A content analysis made it possible to identify the credibility and validity of research findings from each searched study before they could be authorized for inclusion in the systematic review and meta-analysis.

The systematic review and meta-analysis of the chosen relevant studies also revealed the studies’ primary outcomes and risk ratios. The primary outcome of the studies was the number of participants who would show interest in receiving COVID-19 booster vaccination if combined with the influenza vaccine. The hypothesis being tested by the researchers of such studies was to establish the risk that such participants would have in getting COVID-19 and flu upon receiving a combination of COVID-19 booster and flu vaccines. The average risk ratio of the studies was 0.78, which was calculated based on various statistical tests used for analyzing the data collected by the researchers of such studies at a 95% confidence interval (CI). Such statistical tests included *t*-test, chi-square, analysis of variance (ANOVA), binomial, Wilcoxon signed rank, and one sample median tests. Such risk ratio answers the question in this case by revealing that the participants are at a low risk of the intervention groups of the research studies opting to get vaccinated with a combination of COVID-19 booster and flu vaccines. The intervention groups in all the studies were participants who were informed prior to the study that they would receive a combination of COVID-19 booster and influenza vaccines. Contrarily, the control groups for some studies were participants who were informed that they would only receive the COVID-19 booster vaccine [2,3,4,5,6,7,8,9,10,11,12,13,14,15,16,17,18,19,20,21,22,23,24,25,26,27,28,29,30]. For the remaining studies, the control groups were participants who were left to make a choice between having COVID-19 booster vaccines only or a combination of COVID-19 booster and influenza vaccines [1,6,7,11]. For statistical significance with a *p* < 0.05, a risk ratio should not have a 95% CI that includes 1.0 All the research studies that were used for this systematic review and meta-analysis had risk ratios of between 0.56 and 0.96. Therefore, since the proportion of the research studies with risk ratios less than 1.0 was 100%, it can be deduced that all the studies used in the systematic review and meta-analysis had statistically significant risk ratios. The secondary outcome measures of the studies included the number of participants who had received the COVID-19 booster vaccination, the number of participants who were disinterested in receiving the COVID-19 booster vaccination, and the number of participants who were yet to receive the COVID-19 booster vaccination. None of the analyzed research studies experienced an adverse event that could have resulted in health risks among the participants. The Newcastle-Ottawa Quality Assessment Form for Cohort Studies (NOQAFCS) was used to assess the risk of bias in the included studies. It involved determining the quality of research based on specific criteria and grading the studies using scores between 0 and 9. Research articles that had a high-quality research score closer to 9 had a low risk of bias, while those with a low-quality research score closer to 0 had a high risk of bias. A sensitivity analysis was also conducted after excluding studies with a high risk of bias to predict the outcome of their findings based on certain variables that were related to the research topic. There are two main variables for which data were extracted from the chosen research articles. They include the number of vaccinated participants, which refers to participants who had received both influenza and COVID-19 vaccines, and the sex of the participants, which refers to whether a participant was of a male or female gender. Lastly, no summary measures were established and used during the systematic review and meta-analysis of the chosen relevant research articles.

It was also necessary to determine the variability in the findings of the research studies by establishing their heterogeneity. Heterogeneity is measured using a statistic I^2^ which establishes the proportion of the variance between the observed effect and that due to true effects instead of inevitable sampling errors. The I^2^ for the synthesized research articles was 35%, representing low heterogeneity, implying less variability in the findings of the studies based on the set of effect sizes attributed to the sample of participants they used. The studies were combined by exporting their findings to an analysis document and establishing the disparities in such findings based on their research questions, hypotheses, and objectives. A random-effects model was used in that the authors of this study considered an assumption that individual-specific effects did not have any correlation with the independent variables of the chosen relevant studies. The statistical package that was used for the data analyses of the research studies was Microsoft Excel 2019. Lastly, the study does not yet have a registration number or registry name.

### 2.3. Role of the Funding Source

The current study did not require a funder to achieve the goals and objectives of the systematic review and meta-analysis of the chosen research articles. As such, the study had no funder and was solely undertaken through the efforts of the authors who have been named above. On that account, there was no need for a funder to have a role in overseeing the success of its study design, data collection, analysis and interpretation processes, and the writing of the report.

## 3. Results

Through a search of the three databases for identifying the relevant research articles for the systematic review and meta-analysis, a total of 78 research articles were identified. The number of research articles was relatively smaller due to the confinement of the search to the three databases and on the topic of a COVID-19 booster vaccination. Nonetheless, only 56 research articles were left after the removal of duplicates and these underwent the screening process to determine their bias risk. Out of the 56 research articles that were assessed and screened, only 30 were included in the systematic review and meta-analysis after the remaining 23 research articles were identified as ineligible because they were associated with a low quality of research, which implied that they had a high risk of bias, were not peer reviewed, and only had abstracts rather than full texts. Three full texts were excluded and the reason for their exclusion has been elaborated in the current study in the Results section below. Consider the study selection figure that takes the form of a Preferred Reporting Items for Systematic Reviews and Meta-Analyses (PRISMA) flow chart shown in Figure 3.

The included research studies had a wide range of characteristics, the main being that they all used a randomized controlled trial. The main aspect of such study design was that the studies had a sample of randomly sampled participants, and all the variables except those which were experimented upon were kept constant by the researchers of all the included studies. Furthermore, all the studies involved either empirical or applied research in establishing their findings. The last characteristic is that they either employed a hypothesis or research question that guided their findings and was based on the research topic of interest. Consider the summary Table 1 below, which shows the main characteristics of the study for which data were extracted. A primary assessment of the research studies included in this systematic review and meta-analysis provided meaningful and valid findings related to the research topic. One of the findings for the primary assessment was that combining influenza and COVID-19 booster vaccination can be an effective strategy for increasing the uptake of the latter by populations that are reluctant to do so [1,2,3,4,5,6,7,8,9,13,14,15]. Another finding for the primary assessment was that many populations are highly likely to take influenza vaccines when compared with COVID-19 booster vaccines due to the negative perceptions about the health implications and side effects that may be associated with the latter [18,19,20,21,22,23]. On the other hand, one of the findings for the secondary assessment of the chosen research articles was that the uptake of influenza vaccines was significantly higher among diverse age groups and sexes when such vaccines first came out when compared with COVID-19 booster vaccines [24,25,26,27,28]. Another finding for the secondary assessment of the chosen research articles was that combining influenza and COVID-19 booster vaccines is an effective strategy to increase vaccine uptake, even though there is a need to consider the potential health effects among those who had initially been vaccinated against influenza [10,11,12,13,14,15,16,17,29,30]. As aforementioned, the heterogeneity between the analyzed studies was 35% which implies that most of them had almost similar findings related to the topic of research based on their research questions, hypothesis, and sample sizes. In addition, many of the research studies used had a low risk of bias given that they had a high research quality based on the NOQAFCS assessment undertaken during the article screening process. Consider Table 2 below, which provides a summary of the risk of bias and quality of research of the included articles based on such assessment. From a sensitive analysis of the variables of the included studies, it was clear that there was no significant effect on the dependent variable due to a change in the independent variable based on the hypotheses formulated by the researchers in 16 studies. On the other hand, the remaining 14 studies experienced a significant change in their dependent variables upon experiencing a change in their independent variables. Therefore, the uncertainty in the mathematical models used in such 14 research studies resulted in a significant level of corresponding uncertainty in the research model that guided the data collection and interpretation process of such studies.


**List of Studies Excluded at Full Text Screening Stage.**


Soares, P.; Rocha, J.V.; Moniz, M.; Gama, A.; Laires, P.A.; Pedro, A.R.; Nunes, C.; et al. Factors associated with COVID-19 vaccine hesitancy. *Vaccines* 2021, *9*, 300.Reason for exclusion: Insufficient information on influenza vaccines combination with COVID-19.Machingaidze, S.; Wiysonge, C.S. Understanding COVID-19 vaccine hesitancy. Nat. Med. 2021, 27, 1338–1339.Reason for exclusion: Did not use randomized controlled trail design.Andreadakis, Z.; Kumar, A.; Román, R.G.; Tollefsen, S.; Saville, M.; Mayhew, S. The COVID-19 vaccine development landscape. Nat. Rev. Drug Discov. 2020, 19, 305–306.Reason for exclusion: Did not use randomized controlled trial design.

During the systematic review, it was necessary to calculate the average risk ratio of the articles that were included in the synthesis. The risk ratio compares the risk of occurrence of a health event among two distinct groups of a research sample, preferably the intervention and control groups. As previously mentioned, the average risk ratio of the included articles was 0.78 based on a measure of the precision of 95% CI, which implies that there is a relatively lower risk of the intervention group opting to get vaccinated with the COVID-19 booster vaccination in case it was not combined with the influenza vaccine when compared with the control group. Consider the meta-analysis forest plot for each study and effect estimates with a 95% CI that provides a simple summary of data extracted from the included research articles shown in Figure 4 below. From the 30 research articles included in this systematic review and meta-analysis, the total number of participants included in the 30 studies was 1,267,908 people. They comprised people who had received influenza and COVID-19 booster vaccines, people who had not received such vaccines, those who had only received them, and those who had only received COVID-19 booster vaccines. Lastly, the average *p*-value for all the included research studies was *p* = 0.62, while statistical significance occurred at *p* < 0.05 for all the research studies. A total of 19 out of the 30 research studies, which is equivalent to 63.33% of the total studies which were included in the current systematic review and meta-analysis, had significant *p*-values. In essence, many of the research studies had *p*-values that were far above the statistical significance threshold of 0.05. In this way, many researchers opted not to reject their null hypothesis, given that many obtained a *p*-value greater than *p* = 0.05 from their collected and analyzed data.

Consider Figure 4 above, which shows a meta-analysis forest plot for all the studies included in this systematic review and meta-analysis.

## 4. Discussion and Conclusions

From a synthesis of the chosen research studies, it can be concluded that combining influenza vaccines with COVID-19 booster vaccines can be an effective strategy for increasing the uptake of the latter by populations that have shown reluctance against taking it. Researchers have revealed that a significant number of people who have taken the full dose of COVID-19 vaccines are less likely to take the COVID-19 booster vaccines even though such booster vaccines play a significant role in strengthening an individual’s immunity against future COVID-19 transmission and infections [13,14,15]. It is estimated that only 67% of the fully vaccinated population has shown interest in taking the COVID-19 booster vaccines [10]. The major reason behind the reluctance to take such booster vaccines is the perception that the fully vaccinated populations have about having maximum protection against contracting and experiencing negative health effects associated with the virus [16]. Nonetheless, a combination of influenza and COVID-19 booster vaccines can increase the uptake of the latter by over 56%, according to some of the analyzed research studies [1,2,3,4,5,6,7,8]. One of the major reasons behind such an argument is that many populations have felt accustomed to receiving flu shots on an annual basis and will find it easier to replicate such health action in the case of a combined flu shot with COVID-19 booster vaccines [5]. The Centers for Disease Control (CDC) recommends flu shots to be taken every six months during the beginning of each fall season [6]. Another reason is that flu shots have been attributed to an efficacy percentage of more than 50% in healthy adults, which may increase the perception that many populations have towards the efficacy of COVID-19 booster vaccines when they are combined [16]. Finally, such influenza vaccines are mainly taken by older adults, the target population for COVID-19 booster vaccines. It is estimated that 74% of older adults above 65 years take such vaccines, which is significantly higher compared with just 37% of younger adults aged between 18 and 49 years [3]. Since many older adults are highly likely to take influenza vaccines, combining them with COVID-19 booster vaccines can be an effective strategy to increase their interest in receiving such vaccines.

### 4.1. Limitations and Strengths of the Study

Two limitations are associated with the current study. First, not enough research articles were established regarding COVID-19 booster vaccination from the databases mentioned above. Second, a significant number of literature sources with credible and valid information had to be excluded since they did not entail specifically peer-reviewed research processes necessary for systematic review and meta-analysis. Contrarily, one of the strengths of this study is that it relied only on current research articles written and published within the last two years, which justifies the validity of its findings. Furthermore, the research articles were collected from credible databases considered trusted literature sources. Nonetheless, the sources of bias for this research include selection and publication biases.

### 4.2. Controversies Raised by the Study and Suggestions for Future Research Directions

There is only one controversy that the current study has raised. It is whether combining influenza vaccines and the first or second doses of COVID-19 vaccines can be an effective strategy for increasing the willingness of populations to receive the latter. On that account, future research studies should conduct a systematic review and meta-analysis on combining influenza and the first or second doses of COVID-19 vaccines to increase such vaccines’ uptake. They should also focus on a systematic review and meta-analysis of the health implications of combining influenza and COVID-19 vaccines. In such a way, they cannot build on the findings of this systematic review and meta-analysis but also raise further research questions that can add to the body of knowledge of the research topic of this study.

## Figures and Tables

**Figure 1 vaccines-11-00016-f001:**
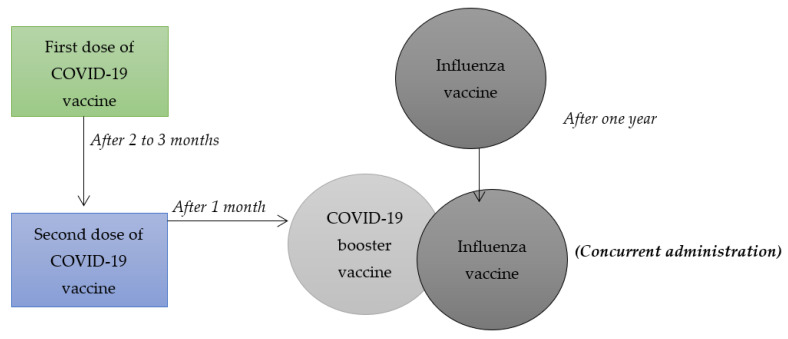
Combining influenza vaccine with COVID-19 booster vaccine.

**Figure 2 vaccines-11-00016-f002:**
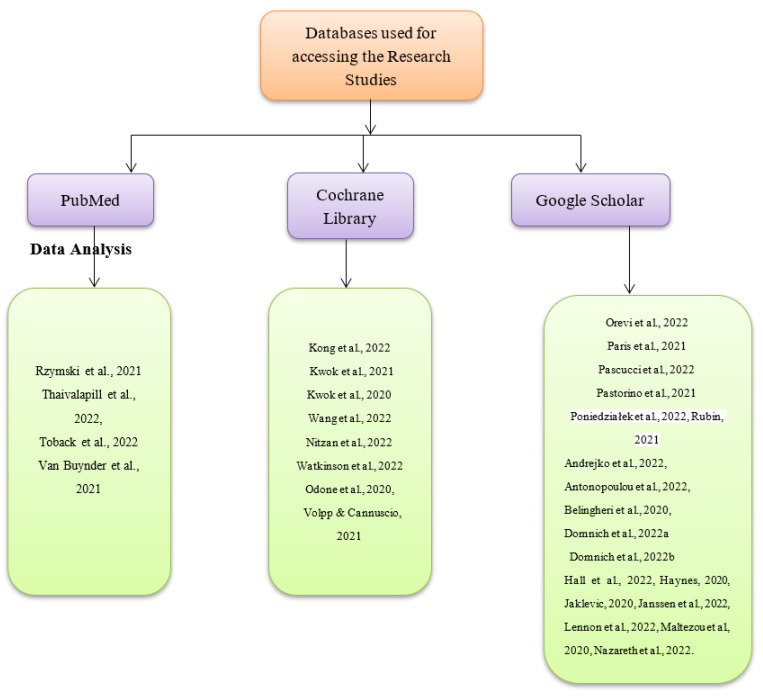
Databases used for identifying the research articles [1,2,3,4,5,6,7,8,9,10,11,12,13,14,15,16,17,18,19,20,21,22,23,24,25,26,27,28,29,30].

**Figure 3 vaccines-11-00016-f003:**
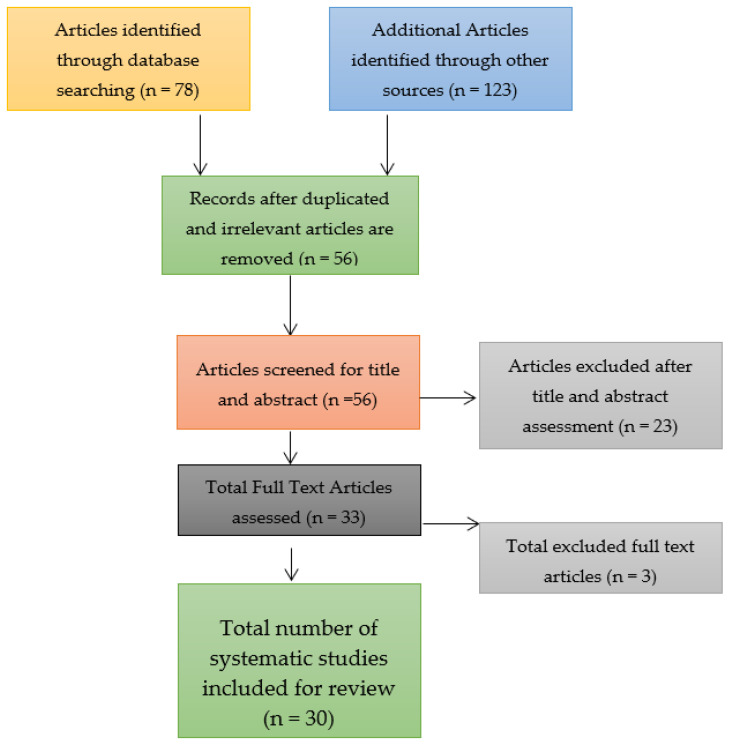
Study selection figure for the systematic review and meta-analysis.

**Figure 4 vaccines-11-00016-f004:**
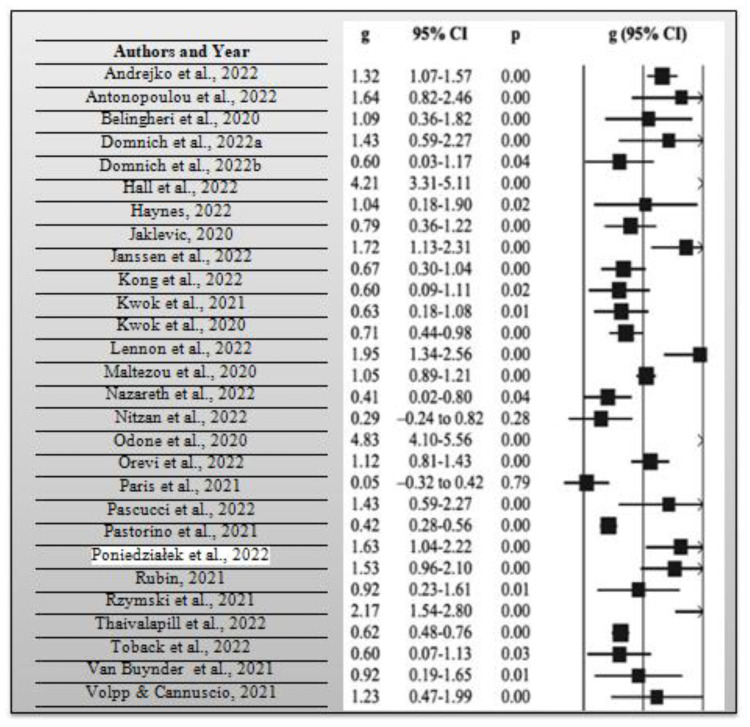
Meta-analysis forest plot for each research study used in this systematic review and meta-analysis [1,2,3,4,5,6,7,8,9,10,11,12,13,14,15,16,17,18,19,20,21,22,23,24,25,26,27,28,29,30].

**Table 1 vaccines-11-00016-t001:** Main characteristics of the research studies used [1,2,3,4,5,6,7,8,9,10,11,12,13,14,15,16,17,18,19,20,21,22,23,24,25,26,27,28,29,30].

	Main Characteristics	
Research Studies	Type of Research	Use of Hypotheses or Research Questions
Andrejko et al., 2022	Empirical	Hypotheses
Antonopoulou et al., 2022	Applied	Research Questions
Belingheri et al., 2020	Applied	Hypotheses
Domnich et al., 2022a	Applied	Research Questions
Domnich et al., 2022b	Empirical	Hypotheses
Hall et al., 2022	Empirical	Research Questions
Haynes, 2022	Empirical	Research Questions
Jaklevic, 2020	Applied	Research Questions
Janssen et al., 2022	Applied	Hypotheses
Kong et al., 2022	Empirical	Hypotheses
Kwok et al., 2021	Empirical	Research Questions
Kwok et al., 2020	Applied	Research Questions
Lennon et al., 2022	Applied	Research Questions
Maltezou et al., 2020	Empirical	Hypotheses
Nazareth et al., 2022	Applied	Hypotheses
Nitzan et al., 2022	Empirical	Research Questions
Odone et al., 2020	Applied	Hypotheses
Orevi et al., 2022	Empirical	Research Questions
Paris et al., 2021	Applied	Research Questions
Pascucci et al., 2022	Applied	Hypotheses
Pastorino et al., 2021	Applied	Research Questions
Poniedziałek et al., 2022	Empirical	Research Questions
Rubin, 2021	Applied	Research Questions
Rzymski et al., 2021	Empirical	Research Questions
Thaivalapill et al., 2022	Applied	Hypotheses
Toback et al., 2022	Empirical	Research Questions
Van Buynder et al., 2021	Empirical	Hypotheses
Volpp and Cannuscio, 2021	Applied	Research Questions

**Table 2 vaccines-11-00016-t002:** Risk of bias assessment [1,2,3,4,5,6,7,8,9,10,11,12,13,14,15,16,17,18,19,20,21,22,23,24,25,26,27,28,29,30].

Authors and Year	Quality of Research Score according to NOQAFCS (0 to 9)	Risk of Bias
Andrejko et al., 2022	7	Low
Antonopoulou et al., 2022	8	Low
Belingheri et al., 2020	6	Low
Domnich et al., 2022a	8	Low
Domnich et al., 2022b	8	Low
Hall et al., 2022	5	Moderate
Haynes, 2022	7	Low
Jaklevic, 2020	8	Low
Janssen et al., 2022	9	Low
Kong et al., 2022	7	Low
Kwok et al., 2021	9	Low
Kwok et al., 2020	6	Low
Lennon et al., 2022	6	Low
Maltezou et al., 2020	7	Low
Nazareth et al., 2022	8	Low
Nitzan et al., 2022	9	Low
Odone et al., 2020	5	Moderate
Orevi et al., 2022	7	Low
Paris et al., 2021	8	Low
Pascucci et al., 2022	8	Low
Pastorino et al., 2021	6	Low
Poniedziałek et al., 2022	9	Low
Rubin, 2021	9	Low
Rzymski et al., 2021	7	Low
Thaivalapill et al., 2022	9	Low
Toback et al., 2022	8	Low
Van Buynder et al., 2021	6	Low
Volpp and Cannuscio, 2021	8	Low

## Data Availability

Not applicable.

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
