# Peer review of "Combining Influenza and COVID-19 Booster Vaccination Strategy to Improve Vaccination Uptake Necessary for Managing the Health Pandemic: A Systematic Review and Meta-Analysis"

_vaccines, 2022, doi:10.3390/vaccines11010016_

Round 1

Reviewer 1 Report

This article is written well, however, some of the points need to be addressed before its publication. I have mentioned all the points below:

1. There are few grammatical mistakes in the manuscript, hence I request authors to check the manuscript carefully before submitting the revised version of manuscript.

2. Don't use COVID-19 twice in the abbreviations.

3. In the introduction, incorporate some information from the article as mentioned below-Making sound public health decisions for the roll-out of COVID-19 vaccines. J Trav Med. 28(4): taab031.

4. This journal follows the numbering format of the references, kindly update the same in the whole manuscript.

5. Figure 3 is not clear. Please provide a good-quality photograph.

6. References are not as per the format of the journal. Kindly update the same in the revisions.

Rest is ok.

Author Response

Dear Esteemed Reviewer

Vaccines

Date: 12/7/2022

RE: The Revisions Made to the Manuscript and Responses to Your Comments and Suggestions

            I hereby express my gratitude for your review of the systematic review and meta-analysis research article on ‘combining of influenza and COVID-19 booster vaccines’. I am also very grateful for your comments and suggestions and wish to gladly inform you in this letter that they have been considered and addressed appropriately within the revised version of the research article that has been attached. For part a of the comments and suggestions that you made, the following amendments have been made in the revised file.

  • A proofreading of the research article has been undertaken and indeed certain grammatical mistakes have been identified and corrected.
  • COVID-19 has now been used only once in the abbreviations.
  • The introduction has been revised to incorporate the effectiveness of the roll-out of COVID-19 vaccines as a sound public health decision.
  • References have now been cited in order and the same has been included in the text in the form, [1], [2], [3] etc.
  • Figure 3 has been deleted and a clearer version has been cut and pasted into the revised version of the research article.

Finally, I would like to express my sincere apologies for the inconvenience caused by the absence of your comments and suggestions. I believe that you will find the revised version of the research article satisfactory for publishing purposes. Thank you!

Yours Faithfully

Prof. Nikolaos Tzenios

Reviewer 2 Report

This paper performs a meta-analysis and a review on the effectiveness of combing COVID-19 booster vaccine with influenza vaccine to increase willingness and motivation to take these vaccines. While the question itself is useful and important, and the analysis is basically valid, the report of its statistical results is not fully effective, and a number of non-accuracies in the interpretation of the results that must be corrected.

In addition, more should be added in the background concerning previous similar studies.

Major comments.

·         There is no literature report on previous similar meta analyses or reviews. Thus the added value of the proposed study is not clear.

·         In the abstract - "The average risk ratio of the included articles…" – risk ratio of what? Please mention briefly what was tested? And likewise "implying that the null hypotheses was not rejected…" – what was the null hypothesis? Although it is the abstract, the reader needs to understand the meaning of the reported risk ratio.

·         In the abstract and in the results section - 0.78% seems like a mistake. Is it really 0.78 percent, or 78 percent? In any case, as far as ratio measurements are concerned, I recommend sticking to the 0 to 1 range, as widely accepted for reporting risk ratios, odds ratios, hazard ratios etc.

·         In "data analysis" – "The primary outcome of the studies was the number of participants who would show interest in receiving COVID-19 booster vaccination if combined with the influenza vaccine…" – please explain how is the risk ratio related to that. What hypothesis is being tested, and how does he risk ratio answer the question in this case? How is the risk ratio obtained - Is it a crude risk ratio or calculated from a model? If so, what model, and is it a multivariate model?

·         In the abstract and in the results section - An average p-value is less informative, as non-significant p-values come from a uniform[0,1] distribution while significant do not. I recommend to report the proportion (or percentage) of significant p-values, across all studies. It will also be useful to describe the distribution of the significant p-values – for instance, were they very far below 0.05? or close?

·         Likewise, in the results section – "As previously mentioned, the average risk ratio of the included articles was… which implies that there is a relatively lower risk of the intervention group opting to get vaccinated with the COVID-19 booster vaccination…" – please report the proportion of risk ratios that were significant. An average of non-effects and effects is rather limited. In addition, it would be useful to know the average or median of the STATISTICALLY SIGNIFICANT risk ratios.

Minor comments:

·         Chart 1 – "After 2 to 3 months" – after what event?

·         In "results" - "Out of the 56 research articles that were assessed and screened, only 30 were included in the systematic review and meta-analysis after the remaining 23…" not clear: 56-23=33, why only 30? (I see that this is clarified in chart 2, but the text needs to be clarified.

Author Response

Dear Esteemed Reviewer

Vaccines

Date: 12/7/2022

RE: The Revisions Made to the Manuscript and Responses to Your Comments and Suggestions

            I hereby express my gratitude for your review of the systematic review and meta-analysis research article on ‘combining of influenza and COVID-19 booster vaccines’. I am also very grateful for your comments and suggestions. I wish to gladly inform you in this letter that they have been considered and addressed appropriately within the revised version of the research article that has been attached. For part a of the comments and suggestions you made, the following amendments have been made in the revised file.

The major and minor comments and suggestions you made following your review process have also been taken into account and addressed appropriately. Kindly observe that the following revisions have been made to the previous research article.

  • The added value of the research study has been included in the introduction section.
  • The average risk ratio has been clarified in the abstract section, and the study's null hypothesis has also been stated in the abstract and the introduction.
  • The risk ratio was indeed 0.78 and not 0.78%. In this regard, revisions have been undertaken as appropriate in the data analysis and results sections and in the abstract.
  • More clarifications on how the average risk ratio was obtained and its significance to the systematic review and meta-analyses has been provided in the data analysis section.
  • The proportion of studies with significant p-values has also been elaborated in the abstract and results section.
  • The proportion of studies with significant risk ratios has also been stated in the results sections in the revised version of the article.
  • Amendments have been made on chart 1 to make it more understandable.
  • Clarifications regarding the three excluded full texts, which have also been mentioned in the results sections have been provided in the results section before chart 2.

Finally, I would like to express my sincere apologies for the inconvenience caused by the absence of your comments and suggestions. I believe that you will find the revised version of the research article satisfactory for publishing purposes. Thank you!

Yours Faithfully

Prof. Nikolaos Tzenios

Round 2

Reviewer 1 Report

The manuscript is improved now. It can be accepted in its present form.

Author Response

Dear Reviewer

Thank you a lot for your positive feedback

Reviewer 2 Report

The authors provided most requested changes. However, some additional improvements are still needed:

1.       Abstract: first added sentences (in highlight) contains too many "based on". Please improve.

2.       In reporting p-values (abstract and results): please specify proportion of SIGNIFICANT p-values (i.e. less then 0.05), rather than the arbitrary non-significance range reported.

3.       When referring to risk ratios:

a.       relative to "control groups" – please provide details what were the control groups in the various studies.

b.       The statistical tests for the risk ratios are still not detailed – what tests/models were used in the various studies?

Author Response

Dear Esteemed Reviewer

Vaccines

Date: 12/9/2022

RE: The Revisions Made to the Manuscript and Responses to Your Comments and Suggestions

            I hereby express my gratitude for your second round of review of the systematic review and meta-analysis research article on ‘combining of influenza and COVID-19 booster vaccines. I am also very grateful for your comments and suggestions and wish to gladly inform you in this letter that they have been considered and addressed appropriately within the revised version of the research article that has been attached. 

Finally, I would like to express my sincere apologies for the inconvenience caused by the absence of your comments and suggestions. I believe that you will find the revised version of the research article satisfactory for publishing purposes. Thank you!

Yours Faithfully

Prof. Nikolaos Tzenios
